# Regulation of Hedgehog signaling Offers A Novel Perspective for Bone Homeostasis Disorder Treatment

**DOI:** 10.3390/ijms20163981

**Published:** 2019-08-16

**Authors:** Wen-Ting Lv, Dong-Hua Du, Rui-Juan Gao, Chun-Wei Yu, Yan Jia, Zhi-Feng Jia, Chun-Jie Wang

**Affiliations:** 1College of Veterinary Medicine, Inner Mongolia Agricultural University, Hohhot 010018, Inner Mongolia, China; 2Department of Veterinary Medicine, College of Animal Science and Technology, Hebei North University, Zhangjiakou 075131, Hebei, China

**Keywords:** hedgehog, RANKL, osteoblast, PTHrP, CREB, NFAT

## Abstract

The hedgehog (HH) signaling pathway is central to the regulation of bone development and homeostasis. HH signaling is not only involved in osteoblast differentiation from bone marrow mesenchymal stem cells (BM-MSCs), but also acts upstream within osteoblasts via the OPG/RANK/RANKL axis to control the expression of RANKL. HH signaling has been found to up-regulate parathyroid hormone related protein (PTHrP) expression in osteoblasts, which in turn activates its downstream targets nuclear factor of activated T cells (NFAT) and cAMP responsive element binding protein (CREB), and as a result CREB and NFAT cooperatively increase RANKL expression and osteoclastogenesis. Osteoblasts must remain in balance with osteoclasts in order to avoid excessive bone formation or resorption, thereby maintaining bone homeostasis. This review systemically summarizes the mechanisms whereby HH signaling induces osteoblast development and controls RANKL expression through PTHrP in osteoblasts. Proper targeting of HH signaling may offer a therapeutic option for treating bone homeostasis disorders.

## 1. Introduction

The hedgehog (HH) protein family, first identified in *Drosophila*, is a highly conserved secretory glycoprotein family that is now known to play key roles in regulating embryonic development, cell differentiation and proliferation, and tissue homeostasis in all vertebrates [1,2,3,4,5]. The *hedgehog* gene was so named because the *hedgehog* mutation in *Drosophila* resulted in the larvae that resembled frightened hedgehogs [6]. When abnormally activated, HH signaling is closely linked to tumor development and metastasis [7,8,9], and tumor cell-derived HH signaling can induce receptor activator of nuclear factor-κB ligand (RANKL) production in osteoblasts, stimulating osteoclastogenesis and increasing bone resorption [10]. The maintenance of bone homeostasis, through the balancing of osteoblast-mediated bone formation and osteoclast-mediated bone resorption, is essential, and the osteoprotegerin (OPG)/RANK/RANKL axis acts as a key regulatory mechanism controlling the differentiation and activity of both osteoblasts and osteoclasts. Both OPG and RANKL can be produced by osteoblasts, with RANKL signaling through RANK expressed on pre-osteoclasts to drive their maturation, thus bone resorption, whereas OPG can also bind RANK, competing with RANKL to block osteoclast induction and inhibit bone resorption [11,12,13,14,15,16]. HH signaling in mature osteoblasts has also been shown to induce parathyroid hormone related protein (PTHrP) expression, which further up-regulates RANKL expression, increasing osteoclast maturation and bone resorption [17,18,19]. HH signaling is therefore thought to regulate many key signals upstream of the OPG/RANK/RANKL axis. HH signaling has also recently been found to stimulate bone marrow mesenchymal stem cells (BM-MSC) differentiation into osteoblasts via influencing Runt-related transcription factor 2 (RUNX2) and Osterix (OSX) expression. Both of these transcription factors are key regulators of osteoblast development and consequent bone formation [3,20,21,22,23]. HH signaling thus serves a key dual role in bone homeostasis by regulating bone formation and resorption. With ongoing research on HH signaling, an increasing number of signaling intermediaries and downstream signaling molecules have been found. In this review, we summarize the molecular mechanisms by which HH signaling induces osteoblast differentiation and regulates RANKL expression through PTHrP in osteoblasts. 

## 2. HH Signaling in Mammals

In mammals, three homologous proteins encoded by the *hedgehog* gene are sonic hedgehog (SHH), desert hedgehog (DHH), and Indian hedgehog (IHH) [24,25,26]. SHH expression is widespread in embryonic tissues and plays critical roles in nervous system, limb, and somite patterning [27,28,29], and it also controls the development of the skin, hair follicles, bones, and gastrointestinal tract [30,31,32]. IHH primarily regulates chondrocyte development and endochondral bone formation, and DHH is expressed predominantly in the male reproductive tract and is essential for the maintenance of the male germ line and spermatogenesis [32,33]. The above three HH proteins serve as secreted signaling molecules and signal through the same highly conserved HH signaling pathway. On target cells, HH signal transduction is controlled by smoothened (SMO) and patched (PTCH, with two mammalian isoforms–PTCH1 and PTCH2), which are 7- and 12-pass transmembrane proteins. The PTCH proteins, encoded by the tumor suppressor gene *Patched*, serve as the primarily HH receptors, binding all three HH proteins to negatively regulate HH signaling [34,35,36]. SMO, encoded by the proto-oncogene *Smoothened*, is a G protein-coupled receptor (GPCR) essential for signal transduction [37,38]. In addition, PTCH is enriched on and around primary cilium of the target cell surface, whereas inactive SMO is located on the membrane of intracellular endosomes and maintained away from the cilium [6,39,40]. The primary cilium is an organelle composed of microtubules and protruding from the surface of most mammalian cells, where it serves as a key mediator of HH signaling [41,42,43]. 

In the absence of HH proteins, PTCH remains near the cilium and represses the activity of SMO. As the final downstream effectors of SMO, GLI proteins, which are zinc finger transcription factors (GLI1-3), are complexed with the suppressor of fused (SUFU) and the kinesin family protein 7 (KIF7). SUFU serves as a major GLI inhibitor, with KIF7 serving a more minor inhibitory role [44,45]. The GLI/SUFU/KIF7 complex can recruit casein kinase I (CKI), protein kinase A (PKA), and glycogen synthase kinase 3 (GSK3). PTCH mediates the activation of these kinases at the cilium base, leading GLI to become phosphorylated. This can in turn promote full-length GLI (GLI-F) cleavage into GLI-R, which is a repressive form that undergoes nuclear translocation to suppress HH target genes important for cellular differentiation, migration, and proliferation (Figure 1a) [46,47]. 

When HH proteins are present, HH binding to PTCH removes PTCH from the cilium, relieving its repression of SMO. SMO then migrates from the intracellular endosome to the ciliary membrane [48], where HH induces SMO phosphorylation via CKI and GPCR kinase 2 (GRK2), leading to ciliary SMO accumulation [49]. SMO then undergoes a conformational change from an inactive to active state [50]. Activated SMO can recruit the GLI/SUFU/KIF7 complex, which undergoes translocation via intraflagellar transport (IFT) proteins to the cilial tip [51,52], where it can mediate the dissociation of GLI from SUFU and KIF7 [53]. The released GLI-F remains in its activator form (GLI-A), traveling through IFT into the cytoplasm and undergoing nuclear translocation to drive HH target gene activation (Figure 1b) [54,55]. 

## 3. Regulation of BM-MSC Osteogenic Differentiation by HH Signaling in Mammals

Mesenchymal stem cells (MSCs) serve as pluripotent progenitor cells which arise from bone marrow that can be induced to differentiate into osteoblasts, chondrocytes, and adipocytes. Osteoblasts and chondrocytes are key cells composing the mammalian skeletal system [56,57]. Bone formation relies upon two essential processes: Intramembranous ossification in certain bones (such as craniofacial bones), and endochondral ossification in the majority of bones (such as long bones). During the former process, BM-MSCs condense and differentiate into osteoblasts and form bone directly, whereas the latter process requires a cartilage intermediate. Endochondral ossification consists of several phases, beginning with the development of a cartilage model, followed by primary and secondary ossification center formation and appositional bone growth. Condensed BM-MSC are able to differentiate into osteoblasts and chondrocytes, with the former differentiating and maturing into bone collars in the cartilage periphery, and the latter forming the initial cartilage model [34,58]. In addition to osteoblasts and chondrocytes, the skeleton also contains hematopoietic-derived osteoclasts, which play vital roles in both cartilage and bone resorption and remodeling [59]. The interaction between osteoblasts and osteoclasts activities results in the necessary bone homeostasis between the formation and resorption of bone. 

Numerous studies have found that osteoblast differentiation is regulated by a number of signaling proteins and transcription factors. HH signaling, and primarily SHH and IHH, are key regulators of embryonic bone development and homeostasis (Figure 2) [18,60,61]. Chiang et al. [27] demonstrated that lack of *Shh* in mice can inhibit endochondral ossification, causing vertebrae and limbs to fail to form. Likewise, Yang et al. [6] found that SHH provides key signals necessary for proper limb bud patterning during early embryonic limb development. Work with mutant *Ihh* mice has revealed that these animals exhibit abnormal endochondral bone formation accompanied by a significant reduction in chondrocyte proliferation, with most chondrocytes being of a mature phenotype and a lack of normal osteoblast development, clearly indicating that IHH plays a central role in the bone formation process [62]. In addition, a recent study established *Ihh* limb-deficient mice, revealing IHH to play a key role in mesenchymal cell differentiation in the limbs, as long bones in these animals showed evidence of severe bone dysplasia, with a loss of normal bone structures due a lack of normal osteoblast activity [63]. HH signaling thus plays a key role in governing normal BM-MSC differentiation into osteoblasts in the context of endochondral ossification. 

RUNX2 and OSX are two essential transcription factors which mediate ossification and osteoblast differentiation processes [64,65,66]. RUNX2 was the first described osteoblast-specific transcription factor, and is also known as core binding factor α 1 (CBFA1). OSX is a downstream target of RUNX2, and is an osteoblast-specific zinc finger transcription factor [66,67]. Luo et al. [3] reported that HH signaling promotes osteoblast differentiation of BM-MSCs via up-regulating the expression of RUNX2 and OSX, inhibiting BM-MSC differentiation into adipocytes. Several studies have further demonstrated that in osteoblastic cells treated with SHH, both *Runx2* and *Osx* gene expression were increased, with *Osx* being up-regulated at an earlier time than of *Runx2*, indicating that factors other than RUNX2 also likely regulate OSX expression, with SHH potentially directly targeting and driving OSX transcription in the context of osteoblastogenesis [68,69]. Moreover, Shimoyama et al. [70] have found that IHH up-regulates the expression and osteogenic action of RUNX2 through GLI2 and consequently stimulates the osteoblastic differentiation of BM-MSCs. Accordingly, GLI2 deficiency in BM-MSCs can significantly inhibit IHH-induced osteoblast differentiation. In addition, IHH treatment or GLI2 overexpression also up-regulates the expression and function of RUNX2. In this study, co-immunoprecipitation further demonstrated that GLI2 and RUNX2 physically interact with one another, suggesting that GLI2 plays a critical role in IHH-induced osteoblastogenesis via regulating RUNX2. Notably, Tu et al. [71] confirmed by histological and molecular analyses that overexpression of RUNX2 in the endochondral skeleton promotes ossification in *Runx2*-null embryos, whereas the same is not true in *Ihh*-null embryos. Therefore, it is possible that osteoblastogenesis in response to IHH necessitates additional factors beyond RUNX2. Other studies have shown that HH family members up-regulate expression of bone morphogenetic proteins (BMPs) in several tissues and cells, and BMPs are pleiotropic cytokines can control osteoblast differentiation and bone formation [72,73,74]. It is thus also likely that IHH achieves its osteogenic action through up-regulation of BMPs expression. In summary, the mechanisms by which HH induces BM-MSC osteoblast differentiation remain poorly understood and necessitate further clarification. 

## 4. Activation of HH Signaling Is Able to Increase Osteoblast Activity

HH signaling is induced during mammalian development and is involved in skeletal formation. During skeletogenesis, HH proteins are produced by differentiating mesenchymal cells, where SHH and IHH drive or maintain cell differentiation programs, regulating cartilage and bone formation, and they are particularly essential for osteoblast generation [2,75]. In addition, HH signaling is involved in homeostatic osteoblast activity and in the regulation of bone remodeling [17]. Indeed, several studies have found that activation of HH signaling can increase osteoblastogenesis and osteoblast activity in cell lines from rodent systems [70,72]. Using a murine bone fracture model researchers demonstrated SHH is activated in osteoblasts at the remodeling site of the fracture and modulates the proliferation and differentiation of osteoblasts [76]. Matsumoto et al. [77] also found that SHH is detected in osteoblasts at the callus remodeling stage in young fractured mice, and it stimulates significantly more alkaline phosphatase (ALP)-positive osteoblast formation. Moreover, Baht et al. [78] found that HH pathway activation in osteoblasts leads to increased matrix deposition at the site of the fracture in mice, and then in vitro osteogenic culture systems further confirmed this pathway activation enhances osteoblast activity. Thus, this suggests that the effect of HH pathway regarding fracture repair is mainly on osteoblasts. 

## 5. Activation of HH Signaling in Mature Osteoblasts Enhances Bone Resorption via Up-Regulation of RANKL

A recent study has shown increased HH signaling activity in mature murine osteoblasts in vivo can indirectly enhance the proliferation and differentiation of osteoclasts [17]. This study further noted that in bones of postnatal mice, HH signaling gradually decreased as osteoblast differentiation and maturation progressed. Up-regulating HH signaling specifically within osteoblasts enhanced the ossification process, however, the resultant bone was highly fragile with a high degree of porosity, and there was clear evidence of bone resorption that eventually led to severe osteopenia in mutant mice. Thus, these lines of evidence suggested that HH signaling in mature osteoblasts drove both bone formation and resorption, with resorption being the dominant process. This may be because enhancement of HH signaling can simultaneously drive osteoblast proliferation as well as osteoclast generation via promotion of RANKL up-regulation, given its central role as an inducer of osteoclastogenesis and osteoclastic function [79,80], thereby accelerating bone resorption. 

### 5.1. HH Signaling Promotes the Expression of RANKL through PTHrP

Accumulating evidence indicates that HH signaling regulates RANKL expression in osteoblasts in a manner inextricably linked to PTHrP activity [6,10,81]. PTHrP is a paracrine regulator secreted by periarticular chondrocytes and osteoblasts [82,83,84], while its receptor is mainly expressed on osteogenic cells [85,86,87]. PTHrP binding to its cognate receptor drives endochondral bone development and the maintenance of the adult skeleton [85,88,89]. In line with previous findings [90], Mak and colleagues [17] reported that PTHrP controls HH-mediated regulation of RANKL expression, promoting RANKL expression via activating multiple downstream cascades. In addition, they further found that activation of SHH promoted enhanced expression of both RANKL and PTHrP in skull osteoblasts from normal mice, but not in those from *PTHrP*-knockout mice. PTHrP thus serves as critical downstream target of HH signaling in osteoblasts to up-regulate RANKL expression, as has been confirmed by other studies [91,92]. 

RANKL belongs to the tumor necrosis factor (TNF) superfamily, and is secreted by osteoblasts and functions as a major regulator of osteoclast differentiation [93,94]. RANKL binds to the receptor RANK on the surface of pre-osteoclasts and mature osteoclasts and facilitates their activation and differentiation and increases bone resorption [95,96]. In a previous study, Mak and colleagues [17] determined that when HH signaling was specifically enhanced in murine osteoblasts, this led to elevated RANKL and PTHrP expression. Interestingly, the expression of these factors rose more quickly than did osteoblast marker up-regulation, indicating that their expression is independent of such osteoblast differentiation in response to HH pathway activation. In summary, in murine osteoblasts HH signaling can drive osteoclastogenesis and bone resorption via PTHrP-mediated modulation of the RANKL to OPG expression ratio. As Vega et al. [97] and Tat et al. [98] have previously described, this RANKL/OPG ratio is an important signal that reflects the bone remodeling environment. A low ratio suggests an environment that favors bone formation, while a high ratio favors bone resorption. 

### 5.2. The Molecular Mechanisms by which PTHrP Regulates RANKL Expression

Research using *Ptch*-knockout mouse models has revealed that enhanced HH signaling in mature osteoblasts ultimately results in elevated osteoclast activity, decreased bone mass, and osteoporosis. The authors of these studies speculated abnormal skeletal phenotype was closely related to PKA and cyclic 3′, 5′-adenosine monophosphate (cAMP) responsive element binding protein (CREB)-mediated RANKL up-regulation [17]. Similarly, Park et al. [99] also proposed that the intracellular cAMP/PKA/CREB cascade is a key signaling pathway important for PTHrP-induced RANKL expression in murine osteoblasts, while CREB downstream of this pathway may be a central regulator of RANKL expression. PKA is made up of 2 regulatory (R) and 2 catalytic (C) subunits, stimulating bone formation owing to its role in osteoblastic cells, enhancing their proliferation and differentiation [100,101,102]. PTHrP binds to its receptor on osteoblastic cells, and the activation of the PTHrP receptor in turn causes the activation of adenylate cyclase (AC) on the surface of osteoblasts [86,99]. Activated AC begins to convert ATP to cAMP. The resultant cAMP accumulates within cells and is able to bind the R subunits of PKA, leading the C subunits to be released. Once activated, these C subunits are able to enter the nucleus and phosphorylate CREB [103,104]. Phosphorylated CREB then binds distal *RANKL* enhancer regions in order to promote elevated expression of RANKL (Figure 3a) [105,106]. The cAMP/PKA/CREB signaling cascade is thus initiated by PTHrP, whereas PTHrP itself is regulated by HH signaling. These signaling molecules thus constitute a signaling axis that regulates RANKL expression in osteoblasts. 

Recent work suggests that in murine osteoblasts, PTHrP-induced RANKL up-regulation is dependent upon CREB and nuclear factor of activated T cells (NFAT) cooperating to bind the promoter region of *RANKL* promoter, such that a reduction in NFAT expression impairs this up-regulation. However, NFAT knockdown did not impair the PTHrP-induced up-regulation of CREB, suggesting it is only essential for up-regulating RANKL in response to PTHrP [99,107]. The NFAT transcription factor family is composed of 5 members (NFATC1-C4 and NFAT 5), each of which undergo specific phosphorylation events that dictate their activity. In the absence of stimulation, NFAT remains in the cytosol wherein it is heavily phosphorylated, whereas upon stimulation high levels of intracellular calcium release mediated activation of the Ca^2+^-dependent phosphatase calcineurin (CN), which dephosphorylates NFAT, leading it to translocate to the nucleus wherein it can regulate gene expression [108,109]. Therein NFAT can cooperate with activated CREB to mediate the binding and transactivation of the *RANKL* promoter in response to PTHrP (Figure 3b). Consistent with this, Takami and his coworkers [110] determined that the calcium ionophore ionomycin can induce RANKL and OPG upon stimulation of osteoblasts, with the former being up-regulated twice as readily as the latter, suggesting that intracellular calcium is capable of increasing the RANKL/OPG ratio, thus driving osteoclastogenesis. We therefore speculate that NFAT overexpression may be a key mediator of intracellular calcium-mediated RANKL expression in osteoblasts, thus driving osteoclast development. 

PTHrP is a key factor responsible for raising circulating calcium levels in the blood, leading to the development of hypercalcemia in certain contexts in addition to playing normal roles in many physiological processes [111,112,113,114]. Increasing intracellular calcium levels in osteoblasts arise due to the influx of extracellular calcium through membrane channels, and/or through the activity of calcium-sensing receptors (CaSR) [115,116,117]. Once these levels of calcium rise to a particular threshold, this is sufficient to drive CN-mediated NFAT activation and consequent changes in gene expression. Interestingly, of the 5 known NFAT transcription factors, Lee et al. [107] observed only NFATC1 and NFATC3 expression in osteoblasts, and they were able to demonstrate that NFATC3 serves as a target gene of NFATC1 in the RANKL regulatory pathway, such that NFATC1 is necessary to mediate NFATC3 up-regulation and activation, whereas overexpressing both NFATC1 and NFATC3 can induce increased expression of RANKL. Park et al. [99], in contrast, proposed that NFATC1 and NFATC3 both bound to the promoter of *RANKL* in murine osteoblasts, with NFATC1 binding being dependent on its interactions with CREB. 

When murine bone marrow cells and primary osteoblasts are cultured together, then treating osteoblasts using ionomycin can drive elevations in intracellular calcium levels, inducing RANKL expression and consequent osteoclastogenesis in the bone marrow cells in a dose-dependent fashion. There was also an initial and unexpected dose-dependent increase in OPG in primary osteoblasts following ionomycin addition, although this expression then decreased over time [110]. This suggests that cytosolic calcium plays a key role in regulating the expression of RANKL, and the ability of RANKL to drive osteoclastogenesis is evidently dominant over the ability of OPG to inhibit this process in a co-culture system. This may in part be due to OPG being a secreted molecule whereas RANKL is a cell-surface molecule [118,119,120,121], potentially leading the former to be more readily diluted than the latter and thus allowing RANKL signaling to dominate and drive osteoclast differentiation. Further, Takami et al. [110] determined that direct protein kinase C (PKC) resulted in comparable outcomes as did increasing intracellular calcium levels, with increases in both RANKL and OPG expression and osteoclastogenesis in a co-culture setting, while PKC inhibitors produced the opposite phenotype. These findings thus clearly reveal a role for PKC downstream of calcium signaling as a mediator of osteoblast RANKL expression. PKC is well-known to act as a kinase central to many fundamental processes within cells [122]. Shin et al. [123] found PKCβ to both induce and activate NFATC1, thereby offering a mechanism whereby it can drive RANKL expression and osteoclastogenesis. These studies thus provide clear evidence that PTHrP, together with PKC signaling, serve as key regulators of osteoblast RANKL expression through their connection to intracellular calcium levels. The calcium/PKC/NFAT signaling axis is regulated by PTHrP as a means of controlling RANKL levels in osteoblasts (Figure 3c).

## 6. Pharmacologic Management of the OPG/RANK/RANKL Axis

Normal bone is always undergoing remodeling in which osteoclast-mediated bone resorption is balanced by osteoblast-mediated bone formation. In addition, bone remodeling also helps to heal injured bones. When this balance is disrupted, bone homeostasis disorders occur, including the most common such disease—osteoporosis. With ongoing research on bone cell biology, specific targets for regulating bone homeostasis have been revealed, which are based on approaches aimed at promoting bone formation or inhibiting bone resorption. To date, the drugs used to improve bone mass mainly include bone-forming agents and bone resorption inhibitors. The former, including parathyroid hormone peptides and strontium ranelate, increase osteogenesis by enhancing osteoblastic activity, proliferation, survival and differentiation, while the latter, including bisphosphonates, selective estrogen receptor modulators, and denosumab, inhibit osteoclastogenesis by reducing osteoclastic activity, survival, and differentiation [124]. As previously mentioned, the OPG/RANK/RANKL axis acts as a key regulatory mechanism maintaining bone homeostasis to prevent bone loss and ensure a normal bone turnover. Thus, as an important target of drug development, manipulation of the RANKL signaling system has attracted increasing attention and research. Currently, denosumab is the only RANKL-targeted therapeutic drug, providing a new treatment option for osteoporosis. This fully human monoclonal antibody is able to bind RANKL with high specificity and affinity and block RANKL-RANK interactions. As a result, osteoclast maturation, function and survival are inhibited, bone resorption slows, and bone mass increases [125]. Numerous studies have confirmed that denosumab markedly increases bone mineral density, with an associated reduction in the risk of fractures [125,126], and that it also decreases the expression of specific markers of bone resorption in postmenopausal women [127]. Therefore, the good application value of denosumab and its unique mechanism of action differ substantially from other bone resorption inhibitors, thus providing meaningful indications for the screening and development of other signaling pathway-targeted drugs. 

## 7. The Effects of HH Signaling on Skeletal Homeostasis Remodeling and Repair

As shown by these previous studies, HH signaling plays a central role in the regulation of bone formation, repair, and homeostasis. Kashiwagi et al. [128] previously found that by locally administering a Smoothened agonist in a murine bone fracture model, they were able to enhance the rate of callus formation in a manner associated with enhanced chondrocyte proliferation in cartilaginous regions and enhanced osteoblastogenesis in bony regions, thus providing direct evidence that enhancing HH signaling can enhance bone repair. In another report, Lee et al. [129] determined that combining a Smoothened agonist with the osteoinductive Nel-like protein-1, they were able to enhance bone healing in a murine calvarial bone defect model in a manner associated with increased osteoblastogenesis, decreased adipogenesis, and an associated enhancement of bone formation. HH signaling is activated in chondrocytes during the early phases of fracture healing and is known to modulate chondrogenesis. Research using mouse models of fracture repair have further indicated that HH pathway activation in osteoblasts positively influences osteogenesis during the later phases of fracture healing, when osteoblasts are maximally activated so as to enhance matrix production at the site of the fracture to form new bone [78]. This suggests that activation of the HH pathway in osteoblasts is more important for normal fracture healing. As described in previous studies, oxygenated derivatives of cholesterol (oxysterols), which are naturally present in the vascular system and tissues of higher mammals, have important functions in many biological processes including cholesterol homeostasis, apoptosis, platelet aggregation, and sphingolipid metabolism [130], and which also have osteogenic potential [131]. Li et al. [132] demonstrated that a novel semi-synthetic oxysterol analogue, Oxy133, strongly induces osteogenic differentiation in cultured rabbit bone mesenchymal stem cells (rBMSCs) in a manner that increases ALP activity, osteogenic gene marker expression, and mineralization, and that mediates successful healing in a rabbit model of intramembranous bone healing in a manner that induces the osteogenic differentiation of undifferentiated mesenchymal stem cells which accumulate at the site of injury. They further confirmed that the osteogenic potency of Oxy133 both in vitro and in vivo is inhibited by the HH signaling inhibitor cyclopamine, suggesting that Oxy133 activates the HH signaling to promote osteogenesis in the context of intramembrenous bone formation. These findings further support the idea that bone injury will heal as long as undifferentiated mesenchymal cells can differentiate into osteoblasts.

Given that such HH signaling was able to simultaneously promote osteogenesis and suppress adipogenesis, this suggests that controlling HH signaling offers an opportunity to better regulate bone density and treat osteoporosis. Osteoporosis is a very common age-related disease in which bone mass and density are reduced and risk of fracture is increased [133]. Nakamura et al. [69] found that the small molecule HH agonist Hh-Ag 1.7 identified through a high-throughput screening effort strongly activates HH signaling in cultured murine mesenchymal stem cells via activating *Gli1* gene expression and stimulates the differentiation of mesenchymal stem cells into osteoblasts via up-regulating *Osx* gene expression, thereby inducing osteogenesis. As such, Hh-Ag 1.7 may have a great potential for the treatment of bone fractures, especially in patients with osteoporosis or the elderly. As previously reported, speckle-type POZ protein (Spop), an E3-ubiquitin ligase adaptor protein, is responsible for the ubiquitination and degradation of mammalian GLI proteins [134,135]. Cai and Liu [136] found that the loss of *Spop* causes osteoblast differentiation defects in mutant *Spop* mice, thereby leading to lower bone density and osteopenia. Of note, the loss of *Spop* leads to increased GLI3 repressor (GLI3-R) levels and decreased IHH signaling. In addition, they further revealed that Spop directly targets GLI3-R for degradation. As GLI3-R is known to antagonize IHH signaling during endochondral ossification [137], they concluded that Spop positively modulates IHH signaling to promote bone development and homeostasis remodeling via specifically down-regulating GLI3-R, and that the Spop protein may thus be a potential target of intervention for osteoporosis. 

Osteoarthritis (OA) is a common degenerative disorder of articular cartilage, especially in the aging population and is associated with obesity. Notably during OA, articular cartilage chondrocytes undergo changes in gene expression and phenotype leading to a phenotype similar to that of chondrocyte hypertrophy in the growth plate during skeletogenesis, indicating the central role for chondrocytes in the progression of OA [138]. As important signaling proteins for the growth and differentiation of chondrocytes, HH signaling has also been shown to play a key role during OA development. Using human osteoarthritic samples and OA mice, Lin et al. [139] found that HH downstream targets, including GLI1, PTCH and hedgehog-interacting protein (HHIP), are highly expressed in articular cartilage accompanied by up-regulation of OA markers, including a disintegrin and metalloproteinase with thrombospondin type 1 motif-5 (ADAMTS5), collagen type X α1 (COL10A1), RUNX2 and matrix metallopeptidase-13 (MMP13), confirming that activation of HH signaling is associated with OA. They used transgenic mice in which HH signaling is activated and further suggested that the level of HH signaling activation correlates positively with the severity of OA. In contrast, when this signaling is inhibited through genetic or pharmacological means, the severity of OA in mice is reduced, and which process potentially is mediated by RUNX2 up-regulating ADAMTS5, which is known to be a key mediator of damage in the development of OA. Thus, inhibiting HH pathway may offer an approach to treating or preventing OA development. Ali et al. [140] also proposed that activation of HH pathway leads to changes in OA progression, with the higher levels of GLI-mediated transcriptional activation in chondrocytes linked to the more severity of OA. Using genetically modified mice, they further demonstrated that the level of GLI-mediated transcription correlates positively with the level of chondrocyte-specific cholesterol accumulation and the severity of OA, suggesting that HH pathway positively regulates intracellular cholesterol biosynthesis, while this cholesterol accumulation in turn partially mediates the effect of HH pathway in chondrocytes in OA. Furthermore, cholesterol blockade, such as statin treatment attenuates the severity of OA in the mouse model with HH activation. Thus, these findings provide new insight into the therapy of OA. Together, there is thus clear clinical value in developing a more in-depth understanding of the mechanisms whereby HH signaling governs skeletal homeostasis remodeling and repair. 

## 8. Conclusions and Future Prospects

As highlighted in this review, HH signaling pathway activation progressively decreased over the course of osteoblast differentiation under normal physiological conditions. Pre-osteoblasts can be driven to undergo more rapid proliferation and differentiation in response to HH signaling, ultimately resulting in increases in bone mass. However, when mature osteoblasts are instead exposed to increased HH signaling then this can result in increased PTHrP expression, activating the cAMP/PKA/CREB pathway and leading CREB to bind the promoter region of the *RANKL* gene, while increased intracellular calcium levels simultaneously activate CN and PKC to promote NFAT activation and binding to the *RANKL* promoter. Together, NFAT and CREB are then able to cooperate to promote the expression of RANKL, thereby driving pre-osteoclasts to mature and thus enhancing rates of bone resorption (Figure 3). 

HH signaling, though often regarded as a linear pathway, is in reality quite complex, with extensive crosstalk between different signaling pathways together serving to regulate physiological processes. While the specific molecules that serve to regulate and execute HH signaling responses are incompletely understood and in some cases remain controversial, there is nonetheless clear evidence that HH signaling is a key regulator of BM-MSC differentiation into osteoblasts, in addition to governing embryonic bone development [141,142]. Indeed, several studies have found that pharmacological modulation of HH signaling can control the rate of fracture healing in model systems. Early during fracture repair, *Ihh* and *Ptch1* are both induced, with osteoblasts exhibiting an increase in SHH signaling at the site of the fracture, thereby regulating osteoblast and osteoclast proliferation and differentiation, in addition to controlling local vascularization [76,143,144]. When HH agonists are administered systemically, they have also been shown to enhance bone and vascular healing in murine fracture model systems [145]. This is consistent with a central role for HH early during osteoblast development and in promoting the proliferation of these cells. As such, there is a clear need to better understand the HH signaling pathway so as to leverage it to enhance reparative osteogenesis in therapeutic contexts of fracture healing of skeletal weakness.

Major strides in the understanding of HH signaling have been made in recent years, allowing for a more mechanistic understanding of how this pathway induces osteoblast RANKL expression. When activated in mature osteoblasts, HH signaling can simultaneously promote bone formation and resorption, with the latter being more dominant. This occurs due to HH-induced PTHrP expression, which in turn drives RANKL expression and osteoclastogenesis. In these mature osteoblasts, HH signaling thus controls the delicate balance between the development and removal of bone tissue through PTHrP and RANKL regulation. The exact outcomes of this homeostatic process rely on context-dependent cues such as cell type, signal strength, and signal timing. For example, Tomimori Y et al. [146] recently found that soluble RANKL administration was able to rapidly drive bone loss in mice through activation of bone resorption. The authors were then able to use this soluble RANKL to induce a bone loss model in which they could assess the efficacy of inhibitors of bone resorption and other compounds through modulation of the balance between bone loss and bone formation. Work conducted to date thus provides clear evidence that control of the HH signaling pathway may represent an optimal means of tipping the balance of bone homeostasis in a clinically advantageous manner in a wide range of human diseases. 

## Figures and Tables

**Figure 1 ijms-20-03981-f001:**
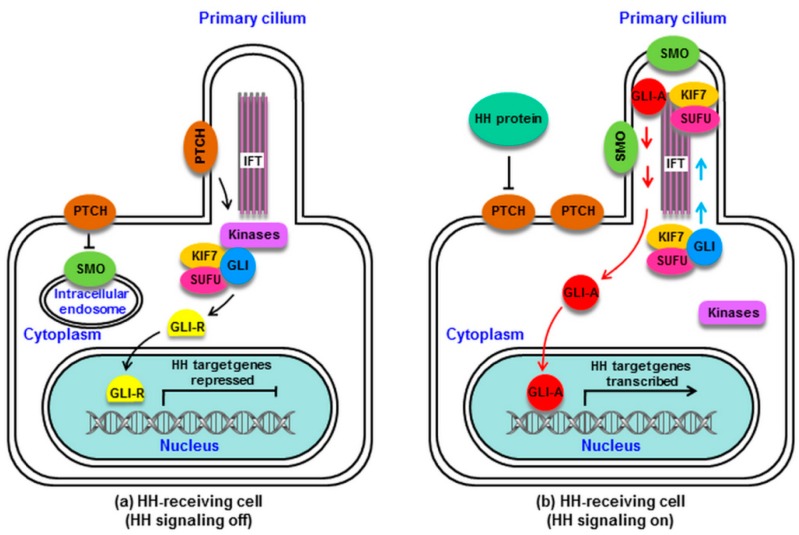
Mammalian hedgehog (HH) signaling. (**a**): Without HH proteins, patched (PTCH) is located at the cilium of the HH-receiving cell, and smoothened (SMO) remains distant from the cilium. PTCH inhibits SMO via preventing its cilial entry. Several kinases (PKA, CKI, GSK3) are recruited by the GLI/suppressor of fused (SUFU)/kinesin family protein 7 (KIF7) complex and activated by PTCH at the base of the cilium. GLI is phosphorylated by these kinases, which promotes its processing into the repressor GLI-R, which undergoes nuclear translocation to block HH target gene transcription. (**b**): HH protein binding to PTCH relieves SMO inhibition. PTCH is displaced from the cilium, and SMO accumulates in the cilium and converts from an inactive to active state. The GLI/SUFU/KIF7 complex is recruited by active SMO, traveling via intraflagellar transport (IFT) to the cilial top. SMO activation results in the detachment of GLI from SUFU and KIF7. GLI remains as an active GLI-A, which is translocated into the nucleus and goes on to induce HH target gene transcription. HH signaling is activated. CKI, casein kinase I; GLI, GLI protein (a zinc finger transcription factor); GLI-A, GLI activator; GLI-R, GLI repressor; GSK3, glycogen synthase kinase 3; IFT, intraflagellar transport protein; KIF7, kinesin family protein 7 (a minor inhibitor of GLI); PKA, protein kinase A; PTCH, Patched; SMO, Smoothened; SUFU, suppressor of fused (a major inhibitor of GLI).

**Figure 2 ijms-20-03981-f002:**
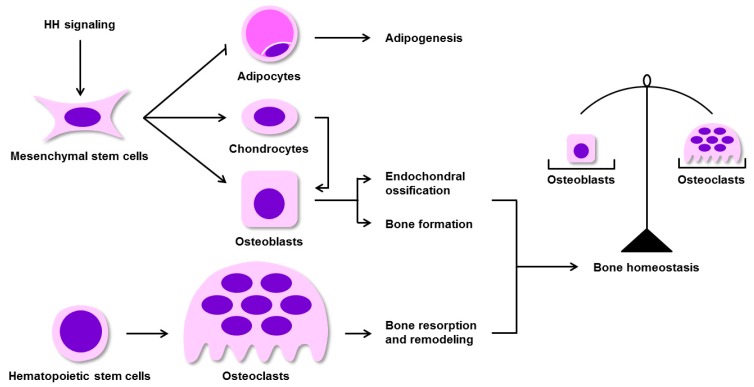
HH signaling is an essential for maintaining bone homeostasis. HH signaling promotes BM-MSC differentiation into chondrocytes and osteoblasts (two cell types involved in endochondral ossification), and inhibits BM-MSC differentiation into adipocytes (associated with adipogenesis). Osteoblasts mediate bone formation, whereas the hematopoietic system-derived osteoclasts drive bone resorption and remodeling. The balance of osteoblast and osteoclast activities is essential for the maintenance of bone homeostasis. BM-MSCs, bone marrow mesenchymal stem cells.

**Figure 3 ijms-20-03981-f003:**
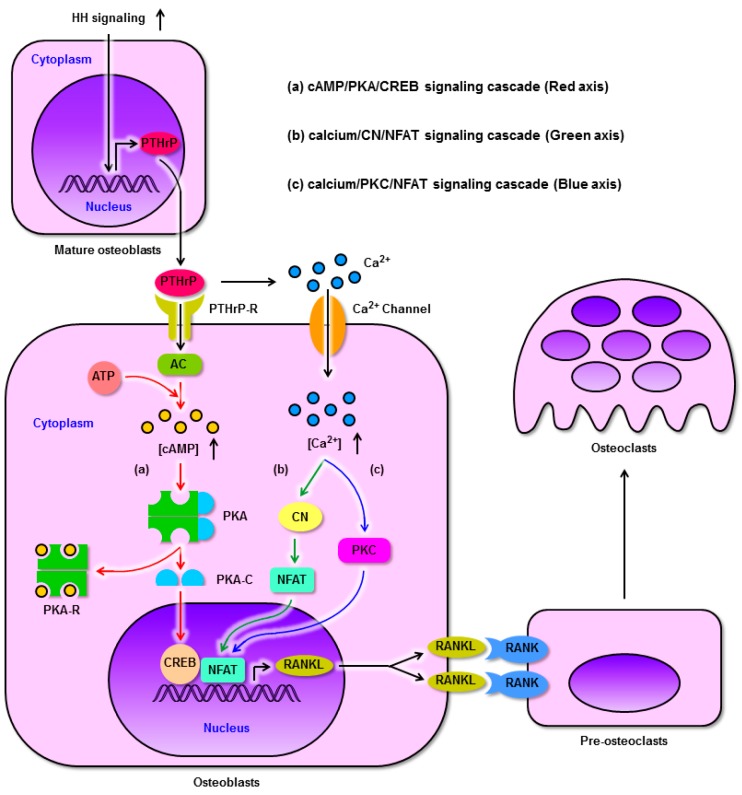
HH signaling regulates RANKL expression through PTHrP in osteoblasts and promotes osteoclast differentiation. (**a**): Enhanced HH signaling up-regulates the expression of PTHrP in mature osteoblasts, after which PTHrP binds its receptor on the osteoblast surface and activates AC. Activated AC converts ATP to cAMP. cAMP accumulates within cells and binds PKA-R, freeing the PKA-C subunits which undergo nuclear translocation and phosphorylate CREB. Subsequently, phosphorylated CREB binds to the *RANKL* promoter. (**b**) and (**c**): Meanwhile, PTHrP elevates intracellular calcium concentrations to activate both CN and PKC, in turn promoting the binding of NFAT to the *RANKL* promoter. As a result, CREB and NFAT cooperatively induce RANKL expression. Finally, RANKL binds RANK on the pre-osteoclast surface, driving osteoclastogenesis. AC, adenylate cyclase; cAMP, cyclic AMP; CN, calcineurin; CREB, cAMP responsive element binding protein; NFAT, nuclear factor of activated T cells; PKA, protein kinase A; PKA-C, PKA catalytic (C) subunit; PKA-R, PKA regulatory (R) subunit; PKC, protein kinase C; PTHrP, parathyroid hormone related protein; PTHrP-R, PTHrP receptor; RANK, receptor activator of nuclear factor-κB; RANKL, RANK ligand.

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
