# Peer review of "Regulation of Hedgehog signaling Offers A Novel Perspective for Bone Homeostasis Disorder Treatment"

_ijms, 2019, doi:10.3390/ijms20163981_

Round 1
Reviewer 1 Report
In their manuscript “Regulation of HH signaling offers a novel perspective for bone disease treatment” the authors Lv, Du, Gao et al. summarize literature data on hedgehog signaling pathways in osteoblasts and their effects on bone development and homeostasis. They mainly focused on following topics: 1. Hedgehog mediated regulation of osteoblast differentiation from bone marrow mesenchymal stem cells and 2. Hedgehog mediated regulation of RANKL expression in mature osteoblasts initiating osteoclastogenesis.
The manuscript provides a good overview of the literature and introduces newcomers very well to the topic of Hh and osteoblasts. The clearly illustrated figures show the signal pathways described in detail in the text at a glance.
However, to publish this review article in the International Journal of Molecular Sciences, the following questions/suggestions should be addressed:
In order to build bones, not only differentiation of osteoblasts but also activity of osteoblast is necessary. Is anything known about hedgehog initiated signaling pathways having an impact on osteoblast activity during bone formation? If information is available, an extra paragraph should be added.
The paragraph “HH signaling in mammals” is very detailed, but it remains unclear if the information given is also true for osteoblasts.
If the HH-producing cell is described (e.g. in figure 1), information about the secretion of hedgehogs are needed as well. It is not enough to mention dispached only, details about scube and sheddases are missing.
Particularly in paragraph 5, the authors describe HH signaling in callus formation and osteoarthritis. In these processes mainly chondrocytes and, to a lesser extent, osteoblasts are involved. The authors should clearly differentiate between the cell types and specify their statements.
The same applies to the term “bone disease treatment”. Which bone diseases are meant? Please specify!
Minor points:
In the title, Hedgehog should not be abbreviated.
Author Response
Dear reviewer and editor,
Thank you for your comments concerning our review. We have studied comments carefully and have made correction which we hope meet with approval.
As young researchers, our knowledge is still very limited, so your comments and guidance give us a great help. And we believe that your suggestions will be a new subject for future work. We will continue to work hard.
Thanks again.
Yours sincerely,
Wenting Lv
Response to Reviewer 1 Comments
In order to build bones, not only differentiation of osteoblasts but also activity of osteoblast is necessary. Is anything known about hedgehog initiated signaling pathways having an impact on osteoblast activity during bone formation? If information is available, an extra paragraph should be added.
1. According to the reviewer’s suggestion, an extra paragraph has been added on Page 5 Line 170-185.
The paragraph “HH signaling in mammals” is very detailed, but it remains unclear if the information given is also true for osteoblasts.
2. According to the reviewer’s suggestion, some information has been added on Page 5 Line 170-185.
If the HH-producing cell is described (e.g. in figure 1), information about the secretion of hedgehogs are needed as well. It is not enough to mention dispached only, details about scube and sheddases are missing.
3. Page 2 Line 63-66: According to the reviewer’s suggestion, “Initially, after a series of intracellular modifications, the HH proteins are converted into active multimeric forms through assistance from the transmembrane transporter protein Dispatched (DISP), allowing for HH protein secretion from the cell in which it was produced (Figure 1a) [34]” has been deleted.
Page 3 Line 96-97: According to the reviewer’s suggestion, “(a): HH proteins are released from the membrane of a HH-producing cell by DISP” has been deleted.
Page 4 Line 107-108: According to the reviewer’s suggestion, “DISP, Dispatched (a transmembrane transporter-like protein)” has been deleted.
Figure 1 has also been modified accordingly.
Particularly in paragraph 5, the authors describe HH signaling in callus formation and osteoarthritis. In these processes mainly chondrocytes and, to a lesser extent, osteoblasts are involved. The authors should clearly differentiate between the cell types and specify their statements.
4. According to the reviewer’s suggestion, some information has been added on Page 10-11 Line 346-351 and Line 363-365.
In addition, to better match the theme of our manuscript, we have added some information about the effects of HH signaling on bone homeostasis remodeling and repair, especially osteoporosis on Page 11 Line 368-384.
Moreover, “Results from both human and murine research efforts has further confirmed that in individuals affected by osteoarthritis (OA), there is increased HH signaling pathway activation, suggesting that inhibiting this pathway may offer an approach to treating or preventing OA development [123]. In animals serving as models of progressive osseous heteroplasia (POH), research has also provided clear evidence for increased HH signaling in ectopic osteoblasts, and when this signaling is inhibited through genetic or pharmacological means, this reduced heterotopic ossification severity [124,125].” has been deleted on Page 10 Line 312-318 in the previous manuscript.
The same applies to the term “bone disease treatment”. Which bone diseases are meant? Please specify!
5. Page 1 Line 25: According to the reviewer’s suggestion, “bone-related diseases” has been changed to “bone homeostasis disorders”.
Page 10 Line 336: According to the reviewer’s suggestion, “skeletal health and disease” has been changed to “skeletal homeostasis remodeling and repair”.
Page 12 Line 412: According to the reviewer’s suggestion, “bone development” has been changed to “skeletal homeostasis remodeling”.
Minor points: In the title, Hedgehog should not be abbreviated.
6. According to the reviewer’s suggestion, the title has been changed to “Regulation of Hedgehog signaling offers a novel perspective for bone homeostasis disorder treatment”.
In addition, additional references ([75], [77], [78], [124], [125], [126], [127], [130], [131], [132], [133], [134], [135], [136], [137], [138], [139] and [140]) have been added. The list of references has also been reordered.

Reviewer 2 Report
Comments and suggestions:
In general, I consider this an interesting, current, well written article however I feel it could be improved if certain recommendations are taken are taken into consideration.
Specific comments
1-The chosen title: “Regulation of HH signaling offers a novel perspective for bone disease treatment” it does not really reflect the broad spectrum of bone diseases (bones can also develop infections or cancer) rather it focuses on those associated with bone density or bone homeostasis. Perhaps a less generic title and more focused on the topic that the reader finds in the text would be more appropriate. Another alternative is to leave the same title but mention in point 5 of how some tumors and infections can cause alteration in the levels of RANKL and/or OPG within the bone stroma.
2-Missing information about the current pharmacological therapy or treatment for bone mass regulation: I consider that each points addressed in the article It has been clear, concatenated and sufficient but a prelude point 5 is required about the treatment that is currently focused on regulation of RANKL and/or OPG (for example the medication carried out with Denosumab) before addressing the benefits of a treatment regulating the HH signaling pathway.
Author Response
Dear reviewer and editor,
Thank you for your comments concerning our review. We have studied comments carefully and have made correction which we hope meet with approval.
As young researchers, our knowledge is still very limited, so your comments and guidance give us a great help. And we believe that your suggestions will be a new subject for future work. We will continue to work hard.
Thanks again.
Yours sincerely,
Wenting Lv
Response to Reviewer 2 Comments
1-The chosen title: “Regulation of HH signaling offers a novel perspective for bone disease treatment” it does not really reflect the broad spectrum of bone diseases (bones can also develop infections or cancer) rather it focuses on those associated with bone density or bone homeostasis. Perhaps a less generic title and more focused on the topic that the reader finds in the text would be more appropriate. Another alternative is to leave the same title but mention in point 5 of how some tumors and infections can cause alteration in the levels of RANKL and/or OPG within the bone stroma.
1. According to the reviewer’s suggestion, the title has been changed to “Regulation of Hedgehog signaling offers a novel perspective for bone homeostasis disorder treatment”.
Page 1 Line 25: According to the reviewer’s suggestion, “bone-related diseases” has been changed to “bone homeostasis disorders”.
Page 10 Line 336: According to the reviewer’s suggestion, “skeletal health and disease” has been changed to “skeletal homeostasis remodeling and repair”.
Page 12 Line 412: According to the reviewer’s suggestion, “bone development” has been changed to “skeletal homeostasis remodeling”.
In addition, to better match the theme of our manuscript, we have added some information about the effects of HH signaling on bone homeostasis remodeling and repair, especially osteoporosis on Page 11 Line 368-384.
2-Missing information about the current pharmacological therapy or treatment for bone mass regulation: I consider that each points addressed in the article It has been clear, concatenated and sufficient but a prelude point 5 is required about the treatment that is currently focused on regulation of RANKL and/or OPG (for example the medication carried out with Denosumab) before addressing the benefits of a treatment regulating the HH signaling pathway.
2, According to the reviewer’s suggestion, some information has been added on Page 10 Line 311-335.
In addition, additional references ([75], [77], [78], [124], [125], [126], [127], [130], [131], [132], [133], [134], [135], [136], [137], [138], [139] and [140]) have been added. The list of references has also been reordered.

Reviewer 3 Report
The submitted review details the role of HH signaling in bone homeostasis and how distinct HH targets (mesenchymal stem cells vs. mature osteoblasts) differentially respond to HH to induce osteoblast formation vs. osteoclastogenesis. The manuscript is well-written, highly detailed, is well-supported by substantial literature and provides substantial insight into the importance of highly-orchestrated HH action.
A major question that was not sufficiently addressed is what regulates HH signaling. While the downstream HH signaling pathways and components were well-described, the upstream regulator(s) of HH was not. Candidate HH regulators may be gleaned from other systems that are subject to HH signaling (e.g., uterus) and discussed for applicability to HH signaling in bone development. These information should be added.
Another area of interest that was mentioned but not well-explained is the role of HH signaling in osteoarthritis. Given that this condition is increased in the aging population and is associated with obesity, an understanding of how and why HH signaling is induced/activated under these conditions may be of therapeutic importance and as a preventative strategy. In particular, given that both aging and obesity are associated with inflammation, would cytokine dysregulation be linked to HH signaling? These information should be added.
Finally, the last paragraph in the review may need to be restructured so that it is not repetitive of the content in other sections. Perhaps this last paragraph can be omitted.
Author Response
Dear reviewer and editor,
Thank you for your comments concerning our review. We have studied comments carefully and have made correction which we hope meet with approval.
As young researchers, our knowledge is still very limited, so your comments and guidance give us a great help. And we believe that your suggestions will be a new subject for future work. We will continue to work hard.
Thanks again.
Yours sincerely,
Wenting Lv
Response to Reviewer 3 Comments
A major question that was not sufficiently addressed is what regulates HH signaling. While the downstream HH signaling pathways and components were well-described, the upstream regulator(s) of HH was not. Candidate HH regulators may be gleaned from other systems that are subject to HH signaling (e.g., uterus) and discussed for applicability to HH signaling in bone development. These information should be added.
1. According to the reviewer’s suggestion, some information has been added on Page 11 Line 351-363 and Page 11 Line 369-384.
Another area of interest that was mentioned but not well-explained is the role of HH signaling in osteoarthritis. Given that this condition is increased in the aging population and is associated with obesity, an understanding of how and why HH signaling is induced/activated under these conditions may be of therapeutic importance and as a preventative strategy. In particular, given that both aging and obesity are associated with inflammation, would cytokine dysregulation be linked to HH signaling? These information should be added.
2. Your suggestion gives us a very interesting writing idea. It needs to be explained that our manuscript focuses on the connections among HH signaling pathway, osteoblasts and bone homeostasis remodeling and repair. In contrast, the occurrence of osteoarthritis is closely related to abnormal changes in articular cartilage and chondrocytes. Thus, considering the theme of our manuscript and the comments made by the three reviewers, we identify bone homeostasis disorders as the main direction of manuscript writing. However, we are not sure whether our ideas are correct or not, so our manuscript has been revised as follows:
(1). On the one hand, according to the reviewers’ suggestion, “Results from both human and murine research efforts has further confirmed that in individuals affected by osteoarthritis (OA), there is increased HH signaling pathway activation, suggesting that inhibiting this pathway may offer an approach to treating or preventing OA development [123]. In animals serving as models of progressive osseous heteroplasia (POH), research has also provided clear evidence for increased HH signaling in ectopic osteoblasts, and when this signaling is inhibited through genetic or pharmacological means, this reduced heterotopic ossification severity [124,125].” has been deleted on Page 10 Line 312-318 in the previous manuscript. In addition, some sentences have been added on Page 10-11 Line 346-365 and Page 11 Line 368-384.
(2). On the other hand, according to your suggestion, some information about osteoarthritis has been added on Page 11-12 Line 385-410.
If unreasonable, we look forward to your guidance. Thank you very much.
Finally, the last paragraph in the review may need to be restructured so that it is not repetitive of the content in other sections. Perhaps this last paragraph can be omitted.
3. Thank you very much for your suggestion. We have carefully considered your suggestion. However, we believe that the addition of the last paragraph facilitates the structural integrity of our manuscript, and also helps readers better understand the content of the whole manuscript. Thus, we kept the last paragraph. If unreasonable, we look forward to your guidance. Thanks again.
In addition, additional references ([75], [77], [78], [124], [125], [126], [127], [130], [131], [132], [133], [134], [135], [136], [137], [138], [139] and [140]) have been added. The list of references has also been reordered.

Round 2
Reviewer 3 Report
I appreciate the authors' careful response to my original comments. I am pleased with the revisions made to address items 1 and 2.
However, I remain convinced that the last paragraph (lines 439-453) should not be part of the Conclusions and may fit better under Section 7 (The effects of HH signaling...).
Author Response
Dear reviewer and editor,
Thank you for your further comments concerning our review. We have studied comments carefully and have made correction which we hope meet with approval.
Your comments and guidance have helped us learn more and also make our manuscript more reasonable. We will be more confident and work harder to carry out our research.
Thanks again, and happy every day.
Yours sincerely,
Wenting Lv
Response to Reviewer 3 Comments (Round 2)
I appreciate the authors' careful response to my original comments. I am pleased with the revisions made to address items 1 and 2.
However, I remain convinced that the last paragraph (lines 439-453) should not be part of the Conclusions and may fit better under Section 7 (The effects of HH signaling...).
According to the reviewer’s suggestion, the sentences on Page 13 Line 439-453 in the previous manuscript have been transferred to Page 12 Line 413-427 in the current revised manuscript.
In addition, the list of references has also been reordered.
